# Identifying the top 10 research priorities for the school food system in the UK: a priority setting exercise

Désirée Schliemann ,[1] Suzanne Spence,[2] Niamh O'Kane,[1] Cheng Choo Chiang,[1] Dilara Olgacher ,[1] Michelle C McKinley,[1] Jayne V Woodside,[1] on behalf of the GENIUS network

[1]Centre for Public Health, Queen's University Belfast, Belfast, UK
[2]Human Nutrition & Exercise Research Centre, Population Health Sciences Institute, Newcastle University, Newcastle upon Tyne, UK

**Correspondence to**
Prof Jayne V Woodside;
j.woodside@qub.ac.uk

## ABSTRACT

**Introduction** The school food system varies widely between schools and across the UK. There is a need to understand evidence gaps in school food research to allow the development, implementation and evaluation of policies and interventions to support children's healthy eating at school. This study aimed to conduct a priority setting exercise to co-produce research priorities in relation to the UK school food system.

**Methods** The James Lind Alliance process informed this priority setting exercise; all key steps engaged a wide range of UK school food stakeholders (including teachers, parents, principals, school governors, policymakers, caterers). An initial online stakeholder survey identified perceived research priorities. In a second survey, stakeholders were asked to rank these priorities. Lastly, an online priority setting workshop with stakeholders elicited the most important research priorities.

**Results** In 2021, school food stakeholders (n=1280) completed the first survey, from which 136 research priorities were identified. In the second survey, participants (n=107) ranked these research priorities regarding their importance. Lastly, 30 workshop participants discussed and reached consensus on the research priorities. After final refinement by the research team, 18 priorities resulted, with the top 10 being related to the provision of free school meals (effectiveness of cost-effectiveness of different levels of eligibility, including universal provision), implementation of policy (including improving uptake) and food standards, issues around procurement, leadership, inequalities, social norms, the eating environment, food culture throughout the school setting and healthy eating.

**Conclusion** The top 10 research priorities were elicited through a rigorous approach, including a wide range of stakeholders across the UK. These should be considered by policymakers, researchers and others to inform research, evidence-based policy development and, ultimately, improve the UK school food system.

## BACKGROUND

Health and dietary intake data suggest that children in the UK do not meet current dietary guidelines, exceeding recommendations for free sugar and saturated fatty acid intake and lacking fibre.[1] Childhood obesity is a major concern in the UK; many factors

### STRENGTHS AND LIMITATIONS OF THIS STUDY

⇒ The study applied a rigorous seven-step priority setting process and involved a wide representation from school food stakeholders across the UK.
⇒ Pupils' concerns and suggestions were reflected in the research questions included in the survey.
⇒ The participants who took part in the study were self-selected and likely had a strong interest in school food.
⇒ Some regions and stakeholder groups were better represented than others.
⇒ No differentiation was made when setting priorities for primary or secondary schools and some of these priorities may be more relevant to one setting than another.

contribute to suboptimal diet and physical activity behaviours. In England, the prevalence of overweight and obesity in 2021/22 was 14.3% and 23.4%, respectively, of children aged 10–11 years. Even higher levels of obesity have been observed in children from the most deprived areas.[2] Childhood obesity has been associated with an increased risk of developing non-communicable diseases later in life.[3–5] The UK Childhood Obesity Plan 2016[6] emphasises the importance of healthy eating in early life and offer recommendations for change. The plan also suggests that improving children's diets requires engagement from schools and others to create long-term, sustainable change. Since lunches eaten at school contribute up to 30% pupils' overall energy intake,[7 8] schools act as an important setting for improving diet quality and reducing health inequalities.[9]

At a global level, the Research Consortium for School Health and Nutrition, a network of over 70 national governments, was established in 2021 and aims to provide better evidence on the 'effectiveness of school feeding programmes' and provide guidance to policy makers on health, nutrition and education

worldwide.[10] In the UK, in 2020, the Generating Excellent Nutrition in UK Schools (GENIUS) network was established and funded by a UK Prevention Research Partnership network grant.[11 12] The network brought together researchers from a range of backgrounds and stakeholders actively involved in school food provision (i.e., local government, catering providers) to collaborate on various activities, with the key objective of developing an improved understanding of the school food system across the UK.[11 12] The 'school food system' refers to everything related to school food, including food education, school food provision throughout the whole school and school day, subsidisation (e.g., free school meals (FSM)), policies and food standards and includes nursery, primary and secondary schools.

School food provision is constantly evolving, for example, Wales and Scotland are in the process of implementing FSM for all primary school children, and some London boroughs are implementing universal FSM in secondary schools, whereas in Northern Ireland it is still entirely means-tested. The recent COVID-19 pandemic required drastic changes to be made to school food provision,[13] highlighting the central importance of the school food system and its agility when required. Due to school closures, replacement schemes had to be established for families who were eligible for and dependent on FSM.[13] In 2021, the National Food Strategy made specific recommendations for the school food system[14]; these included extending FSM eligibility, changes to food education and funding for holiday activities and food clubs available to FSM-eligible children. In response, the government published their food strategy plan in 2022 that describes the desire to 'spark a school food revolution' including the improvement of school food and introduction of a food curriculum.[15]

In this dynamic time for school food, there is a need to understand current research priorities that will help support positive and sustainable changes within the school food system, including informing the next steps for policy change. One of the aims of the GENIUS network was to identify, through working with stakeholders, key priorities for research related to school food. The James Lind Alliance (JLA) has developed a prioritisation process to identify the top research uncertainties/priorities (presented as research questions) that involves all relevant stakeholders, including service users and providers.[16] This process stresses the importance of patient and public involvement to facilitate more relevant and high-quality research. This study aimed to undertake a priority setting exercise, co-produced with stakeholders, to identify the top 10 research priorities within the school food system.

## METHODS

The JLA prioritisation process was followed, which includes seven key stages (figure 1), to elicit the top 10 research priorities.[17] This research was conducted by researchers who were members of the GENIUS network, using its platform to advertise the research and recruit participants for the various stages outlined below.

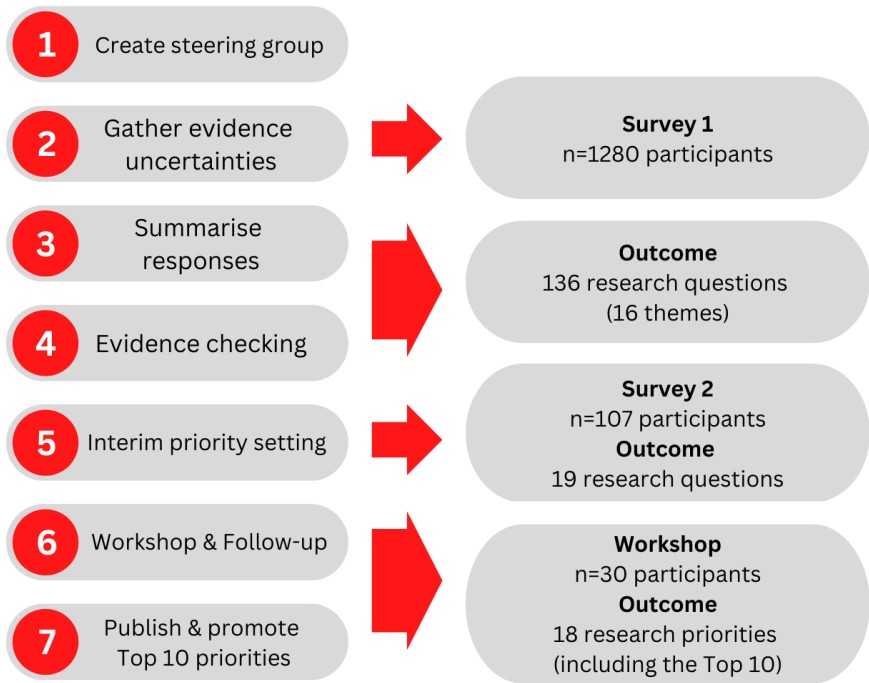

1. Create steering group
2. Gather evidence uncertainties → **Survey 1** n=1280 participants
3. Summarise responses
4. Evidence checking → **Outcome** 136 research questions (16 themes)
5. Interim priority setting → **Survey 2** n=107 participants **Outcome** 19 research questions
6. Workshop & Follow-up → **Workshop** n=30 participants **Outcome** 18 research priorities (including the Top 10)
7. Publish & promote Top 10 priorities

**Figure 1** James Lind Alliance priority setting process.

## Stages 1–4: Create a steering group and gather evidence

A steering group made up of two academics and two practitioners with responsibility for school meal delivery was established to oversee the priority setting exercise and identify potential stakeholders. A broader 'School Food Survey' was developed in collaboration with members of the School Food Review working group; this survey was intended to gather information from school food stakeholders relating to the priority setting exercise described here, and to gather views from caterers on responses to COVID-19 and to explore the views of diverse stakeholders on the school food system more generally; the other survey elements will be published separately. The survey included closed questions (i.e., on region, stakeholder role and school type) and open-ended questions. It was created on Qualtrics XM and promoted via GENIUS membership mailing lists and social media, as well as the networks of members of the School Food Review working group, between 21 June and 21 July and throughout September 2021. Respondents were clearly informed as to what the responses will be used for, that is, identifying areas for future research or to identify or share areas of good practice, and consent was assumed by continuing with the questionnaire after this clear statement of purpose. Survey questions that were relevant for the priority setting exercise are listed below and were regarding areas that participants believed to be important within the school food system. All free-text responses were exported into NVivo V.12 for analysis. The survey responses from primary school stakeholders to the following three questions were initially reviewed and interpreted by one researcher (DS), and turned into a list of indicative research priorities or 'evidence uncertainties' by a second researcher (JVW): "*What specific questions, areas of challenge or opportunity, or topics around the school food system do you think research could help answer or explore?*", "*What do you consider as priority areas for change within the school food system?*" and "*What would an ideal school food system look like from your perspective?*". The list of priorities was then independently checked against responses from secondary school stakeholders by two researchers (JVW and DO) to ensure all responses were captured in the final set of questions. Further evidence uncertainties were identified by another member of the research team (CCC) conducting a scoping review of the school food literature, identifying narrative and systematic reviews in the area and extracting any suggestions for research gaps or need for future work; these were added to the list of priorities.

## Stage 5: Interim priority setting

A second survey, including the research priorities identified at stage 4 of the priority setting process, was created in Qualtrics XM and sent to members of the GENIUS network and to those participants from the first survey who indicated their interest in future-related research. The survey was also promoted through the GENIUS Twitter account to increase responses from stakeholders with an interest in school food, including (vice-)principals, teachers, parents, school governors, researchers, policy makers and others. Participants were provided with written information and asked to provide consent at the beginning of the survey. They were first asked to provide information about their role and region (participants could have more than role and be affiliated with more than one region) and then to rank each priority on a 9-point Likert scale (1=least important and 9=most important). Priorities with a similar theme were grouped together and presented in blocks of priorities to the participants. Blocks of priorities as well as priorities within the blocks were randomised to minimise response bias and differential numbers of responses based on priority order. The survey was live during July and August 2022. All responses were exported into SPSS V.24 for analysis. Duplicate entries were removed and participants with >10% response rate, that is, participants who rated >13/134 research priorities were included in the analysis. It was assumed that participants with fewer responses would not represent meaningful responses. The mean score and standard deviation (SD) was calculated for each research priority.[18] The five highest rated research priorities from each stakeholder group were then combined to be discussed and ranked at the online workshop[19 20] to make sure the top priorities from each stakeholder group were represented. A shortlist of about 18 priorities was considered acceptable for an online workshop.[21]

## Stage 6: workshop

Participants who completed stage 5 of the priority setting process were invited to participate in a workshop that aimed to identify the top 10 priorities for school food research. Furthermore, members from the GENIUS network were invited and primary and secondary schools across the UK were contacted via email to invite teachers and principals with an interest in school food. Prior to the workshop, participants were emailed an information pack, including an information sheet and the list of research priorities/uncertainties. Participants were asked to individually rank the priorities in terms of their importance, from lowest to highest priority, and to complete a short online survey where they were asked about their role within the school food system and to provide informed consent before joining the workshop. The workshop was held online, on Zoom on 29 September 2022. It lasted 3.5 hours and was delivered by four trained JLA facilitators.[21] Participants were first asked to report their highest three and lowest three priorities in small groups (up to eight participants in each group). Each facilitator prepared a PowerPoint slide to reflect which priorities were ranked lower or higher by participants within each small group. Then, participants within the same small groups jointly ranked all priorities according to their perceived importance. Subsequently, the JLA facilitators combined the scores from all four groups which resulted in a combined ranking. Participants were asked to discuss the combined ranking in new small

groups, which was the last opportunity for the order to be changed. A final combined ranking was then produced by the JLA facilitators. The workshop was audio-recorded and seven observers, who kept their microphones and cameras turned off throughout the workshop, made note of the discussions. One researcher (DS) listened to all the recordings and made notes on the justifications for the ranking as well as any suggested rewording of priorities. A postworkshop survey was circulated to all participants for feedback on the process, and provided an opportunity to leave any final comments about the research priorities. The notes from the recordings, observer notes as well as workshop feedback were then combined for each priority and discussed by three researchers from the research team to decide on any potential rewording or merging of priorities. All participants in the workshop received a £10 voucher as a token of appreciation for their time.

### Patient and public involvement

The steering group created included patient and public involvement (PPI) members. After discussion with the Faculty of Medicine Health and Life Sciences Research Ethics Committee, Queen's University Belfast, it was concluded that the questions included in the initial survey, directed at stakeholders, constituted PPI rather than research and did not require ethical approval.

## RESULTS

### Survey (stages 1–4)

The first survey was completed by 1280 school food stakeholders (table 1) who mostly represented England (n=629, 49.1%) and Scotland (n=543, 42.2%), followed by Northern Ireland (n=71, 5.5%) and Wales (n=34, 2.7%). The biggest stakeholder group was parents (n=600, 46.9%) and catering-related staff (n=292, 22.8%) and those associated with a primary school (n=877, 68.5%). All qualitative responses to the three questions were translated into 136 research priorities and grouped together under 16 overarching themes: general (n=8), FSMs (n=9), intake/nutrition (n=16), nutritional standards (n=4), food quality (n=4), cost and financial concerns (n=7), uptake/choice (n=14), menu (n=4), policy (n=18), leadership

**Table 1** Roles and regions of stakeholders involved in all stages of the priority setting process

| | Survey 1 (n=1280) | | Survey 2 (n=107) | | Workshop (n=30) | |
|---|---|---|---|---|---|---|
| | N | % | N | % | N | % |
| **Region** | | | | | | |
| Nationwide | 11 | 0.9 | 6 | 5.6 | 3 | 10 |
| England | 629 | 49.1 | 49 | 45.8 | 18 | 60 |
| Scotland | 543 | 42.4 | 22 | 20.6 | 3 | 10 |
| Wales | 24 | 1.9 | 1 | 0.9 | 3 | 10 |
| Northern Ireland | 71 | 5.5 | 28 | 26.2 | 6 | 20 |
| Other | 6 | 0.5 | 1 | 0.9 | 1 | 3.3 |
| **Role** | | | | | | |
| Teacher/Teaching assistant | 95 | 7.4 | 17 | 15.9 | 3 | 10 |
| Principal | 80 | 6.3 | 3 | | 4 | 13.3 |
| Parent/Carer | 600 | 46.9 | 31 | 29.0 | 1 | 3.3 |
| Pupil | 75 | 5.9 | – | – | – | – |
| Catering-related role | 292 | 22.8 | 6 | | 4 | 13.3 |
| School governor | 20 | 1.6 | 3 | | 1 | 3.3 |
| Local authority practitioner | 29 | 2.3 | 12 | | 7 | 23.3 |
| Researcher/Academic | 6 | 0.5 | 20 | 18.7 | 7 | 23.3 |
| Charity/NGO | 25 | 2.0 | 8 | | 8 | 26.7 |
| Dietitian/Nutritionist | 5 | 0.4 | 4 | | 2 | 6.7 |
| Other* | 50 | 3.9 | 5 | | – | – |
| **School** | | | | | | |
| Primary | 878 | 68.6 | | | 19 | 63.3 |
| Secondary | 437 | 34.1 | | | 12 | 40 |
| Other† | 64 | 5.0 | | | 4 | 13.3 |

*This includes school admin staff and business managers, funders as well as policy- and government-related roles.
†This includes nursery/pre-school, special needs schools and college.
NGO, non-governmental organisation.

**Table 2** Final order of school food priority research questions

| | Full research priority | Short version |
|---|---|---|
| **Research priorities 1–10** | | |
| 1 | What would be the value to pupils' health and well-being of a (i) universal (UFSM) or (ii) extended free school meals (FSM) policy compared with (iii) a means-tested FSM policy and to what extent could such policy changes reduce inequalities? | What is the value to pupils' health and well-being of a UFSM versus means-tested FSM policies, which may have different criteria for eligibility? |
| 2 | How can we improve uptake of FSM for those who are eligible? What factors influence uptake of FSM (e.g., stigma, quality of food, etc.)? What are the most effective interventions to tackle barriers related to FSM uptake and how can these best be implemented at multiple levels? | How can we improve uptake of FSM for those who are eligible? |
| 3 | To what extent is school food valued and prioritised among decision-makers at all levels (e.g., schools, catering, policy, government, etc.)? What are the facilitators and challenges to engaging decision-makers more strongly with school food, and how can these be addressed? What measures of the impact of school food would decision-makers find most important? | What are the facilitators and challenges for decision-makers to engage with and prioritise school food? |
| 4 | How can school food systems best deliver school meals to pupils while keeping within the school food and procurement standards as well as budgets? What are the facilitators and barriers to adhering to these constraints? | How can school food systems deliver school meals to pupils while keeping within the school food standards, procurement requirements and budgets? |
| 5 | How do we implement changes to FSM policies (e.g., extended FSM or UFSM), including consideration of a sustainable funding model, to achieve optimal participation and benefits? | How do we implement changes to FSM policies (e.g., extended or UFSM) to achieve optimal participation and benefits? |
| 6 | How do school food policies and interventions impact on and address inequalities in pupils from different subgroups (e.g., low socio-economic groups, immigrants or ethnic minorities)? What are the barriers and facilitators to allowing pupils from these subgroups to make healthier dietary choices? | How do school food policies and interventions impact on and address inequalities in pupils from different subgroups? |
| 7 | How can policies be developed and implemented (at multiple levels within the school food system) that create a positive food culture within schools? How does this affect pupils' relationships with food? | How can policies create a positive food culture and affect pupils' relationships with food? |
| 8 | All schools are different, how do we overcome the challenges that schools have in implementing an optimal eating environment and making lunchtime a positive social experience for all pupils (including neurodivergent pupils) that is not rushed, gives pupils time to eat and teaches valuable life skills? | How do we overcome the challenges that schools have in implementing an optimal eating environment? |
| 9 | What innovative approaches can be used to make school meals appealing for all pupils and help drive uptake of healthy food consumption in schools (e.g., food presentation, serving options)? | What innovative approaches can be used help drive uptake of healthy food consumption in schools? |
| 10 | What is the impact of all food consumed in school on pupils' health and well-being and to what extent can it reduce inequalities? | What is the impact of food consumed at school on pupils and to what extent can it reduce inequalities? |
| **Research priorities 11–14** | | |
| 11 | How accessible are healthy and unhealthy food options for pupils at school? How can the school food system promote increased consumption of healthy food options over the course of the whole school day? | How accessible are healthy options at school and how can consumption be increased? |
| 12 | How can current food procurement systems be changed in order to promote improved diet quality and sustainability? | How can current procurement systems be changed in order to promote improved diet quality, nutritional status and sustainability? |
| 13 | How can school food system changes help staff, pupils and parents to contribute to the sustainability agenda? | How can school food system changes help staff, pupils and parents to eat more sustainably? |
| 14 | How can behavioural science be used to influence food choices in the school setting (e.g., through the influence of peers)? | What is the influence of peers on pupils' food choice? |

| | | |
|---|---|---|
| **Table 2** | Continued | |
| **Research priorities 15–18** | | |
| 15 | How could inspection requirements/processes ensure health and nutrition is a focus for decision-makers? How should schools measure the impact of their school food provision and any school food policy changes that are implemented? | How could inspection requirements be changed to promote healthy school food among decision-makers? |
| 16 | How is the quality and choice of food that school canteens are able to offer affected by the price they charge? What would parents be willing to pay for a school meal that their child eats? | How is the quality and choice of food that school canteens are able to offer affected by the price they charge? |
| 17 | What is the effectiveness and cost-effectiveness of interventions providing food and activities for pupils during the school holidays? | What is the effectiveness and cost-effectiveness of interventions providing food and activities for pupils during the school holidays? |
| 18 | Do schools have a role to play in identifying food insecurity/food poverty/food shortage in their pupils? How can school staff be supported to achieve this? | Do schools play a role in identifying food insecurity in their pupils? |
| *This refers to parents who are able to pay for school meals. | | |

(n=3), whole school food approach (n=10), canteen/lunch setting (n=6), in-house/external kitchens and staff (n=12), parents (n=5), knowledge, skills and education (n=10), environmental concerns (n=6).

### Survey (stage 5)

Rankings from 107 participants were included in the analysis of the second survey. The majority of participants were affiliated with schools in England (n=49, 45.8%), followed by Northern Ireland (n=28, 26.2%), Scotland (n=22, 20.6%), nationwide (n=6, 5.6%) and Wales (n=1, 0.9%). Parents (n=31, 29%), researchers (n=20, 18.7%) and teachers/teaching assistants (n=17, 15.9%) made up the majority of respondents. After survey completion, the top four or five highest rated questions from each of the following stakeholder groups were extracted: teachers (n=15), parents (n=22), researchers (n=19), local authority (n=10), 'catering staff' (n=5) and 'other' (n=20, including nutritionists, non-governmental organisations, etc). This resulted in 22 different priorities (there was some overlap when the top four or five from each group were merged). There were three questions that were similar and that were merged, which resulted in 19 research priorities that were taken forward to the next stage.

### Workshop (stage 6)

Thirty participants attended the online workshop and ranked 19 questions in terms of their priority. Information about workshop participants is presented in table 1. The postworkshop evaluation survey was completed by 23 participants. The vast majority of participants (n=22/23) were very satisfied/satisfied with the structure and content of the workshop, the workshop facilitation (n=21/23) and the workshop outcomes (n=19/23). Participants appreciated the opportunity to contribute their opinion and listen to a diverse group of participants with different experiences and expertise as well as the facilitation of discussions in smaller groups and the general consensus among participants with regard to the topics of importance. Participants did provide suggestions for the rewording of questions to improve their clarity which were compiled, along with suggestions from the survey, reviewed by three researchers (JVW, MCM, DS) and refinements made resulting in the final set of questions. Feedback related mostly to the terminology used (school meals, school food, FSMs, healthy, nutritious, etc.) and to make sure the questions were inclusive, specific and were not overlapping. Some minor rewording of most questions and combining of two questions in response to this feedback led to a total of 18 research questions (table 2). Three out of the top 10 research questions were regarding FSM, that is, assessing the value of FSM to pupils' health and well-being; improving uptake and implementing changes to FSM policies, including eligibility. Questions around leadership, food and procurement standards, addressing inequalities, establishing a positive food culture and an optimum eating environment as well as increasing healthy food consumption in schools were highly rated. Based on feedback from the workshop participants, a glossary of terms was created to aid the interpretation of the priorities (box 1).

### DISCUSSION

This priority setting exercise, which is the first of its kind in the UK, identified 18 research priorities, presented as research questions (including a list of the top 10) that school food stakeholders consider to be the most important for research related to school food and, ultimately, improve the school food system. The priorities cover a range of areas, including subsidisation of school meals, implementation of school food policy and standards, leadership, inequalities, social norms and healthy eating. Three of the top 10

## Box 1 Glossary of terms

**School food system**
This includes everything related to food in schools. Examples of this include foods available across the whole school day, school food policies, sustainability concerns, procurement, school gardens, food provided at school events, food education in the classroom and activities that integrate with food education, food culture and the environment. It also includes all school food stakeholders (e.g., caterers, suppliers, pupils, teachers, teaching assistants, principals, local government, governors, parents, non-governmental organisations, etc.).

**School food**
This refers to all food consumed in school, for example, snacks, school meals, free school meals (FSM), vending, packed lunch and food purchased outside of school.

**School meals**
This refers to nutritious, well-balanced, appealing, high-quality and culturally acceptable meals that are provided by schools at lunchtime.

**Impact**
By impact we refer to the impact on pupils' dietary intake and educational, attendance as well as health and well-being (e.g., obesity, dental, mental health) as well as economic outcomes (immediate and long-term).

**Free school meals**
FSM refers to free school meals for eligible pupils (which could be for all where universal FSM (UFSM) have been implemented and, where this has not occurred and eligibility is means-tested, for those who meet certain eligibility criteria; the current policy varies across the UK).

**Extended FSM**
This refers to an FSM policy that includes all pupils living in poverty/food insecure households, across the UK (where UFSM have not been implemented).

**Universal FSM**
This refers to FSMs for everyone, across the UK.

priorities are related to the extent, implementation, uptake and effectiveness of FSM provision. In addition, two other priorities related to reducing inequalities among disadvantaged subgroups. There was a lot of discussion about whether every child should receive an FSM (i.e., UFSM) and arguments were made for both extended FSM and UFSM to be further explored. This also depended on the stakeholder region, as, for example, in Wales and Scotland, FSM are being rolled out for all primary school children which is not the case throughout England and in Northern Ireland at this point. It is important to note that the online surveys were conducted during the COVID-19 pandemic and the workshop was conducted at the start of the financial crisis in 2022, when about one in three UK children (4.3 million) were estimated to be living in poverty and 40% of those children suffered from food insecurity.[22] Discussions about inequality were prominent in the workshop, which was likely influenced by the context at this time. Food insecurity is likely to affect children's developmental, well-being

and academic performance[23 24] and ultimately, has been described as a predictor of healthcare utilisation and cost in adults.[25] Therefore, addressing inequalities and supporting the most vulnerable children across the UK to support healthy development and learning was likely a main concern of stakeholders that influenced this exercise. One of the questions relating to school meal prices was rated as of lower priority as participants felt strongly about an extended FSM policy and felt, therefore, that prices should not be of concern for pupils/parents. Also, the argument was made that prices constantly change, especially with the level of inflation at the time. Participants agreed that any change has to start with the support and buy-in from leaders at all levels, especially at a policy level. There was also a lot of discussion around sustainability and environmental impact of the food served in schools; stakeholders agreed that it is important, but argued that providing a nutritious meal to every child had greater priority. Sustainability may also have been an important priority for pupils, who did not participate in the workshop, and therefore 'sustainability' was included in the top 18 priorities, but not in the top 10. Participants also stated that some priorities were not rated highly as they felt they were already being addressed to some extent and they felt that other priorities needed more research attention. This may also have been the case for food education, and, more generally, a whole school approach to food, as included within the National Food Strategy[14] and the resulting government response.[15] It was included within priority 7, although this is implicit, but the more direct food provisioning questions did dominate, which may reflect the current context of financial crisis and the global pandemic.

Other school food priority setting research has been reported from Australia[26] and Canada.[27] For example, researchers in Australia ran a priority setting workshop to identify novel models for school food provision as well as barriers and facilitators towards achieving change. Government support was perceived to be both the top barrier and the top facilitator to changing the school food system, which is consistent with the importance of leadership identified in this UK priority setting exercise. A workshop in Canada focused on the 'perceived needs or issues related to school nutrition policy implementation and evaluation within and outside Ontario'. One of the main outcomes was the need to identify measures to evaluate compliance and enforce the policy, which again is similar to one of the top 10 priorities identified in the UK context.

The list of priorities identified in this study is a guide for researchers and policy makers to inform future funding and research. It is envisaged that research priorities could be combined, for example, sustainability could be addressed together with most other research priorities. The priorities represent general themes/questions which can be adapted depending on the setting, for example,

according to school type and region. This guide will help researchers to develop more specific research questions, based on their setting of interest. Future research should examine intervention effectiveness, and should consider factors such as cost-effectiveness, and the implementation practices, longevity and processes of these programmes, thus evaluating interventions in real-world settings. The priorities can also be used to guide funding agendas; it could be argued that support of research to address these research priorities could ultimately lead to more effective policies around school food and an improved understanding of the UK school food system (the original aim of the GENIUS network).

A strength of this study is the rigorous seven-step priority setting process that was applied, as set out by the JLA, including the facilitation of the workshop by trained JLA facilitators. Also, there was a wide representation from school food stakeholders across the UK included in this process. Pupils were not included in the second survey and the workshop, however, their concerns and suggestions were reflected in the research questions included in the second survey. A limitation of this research is that participants who answered the surveys and participated in the workshop were self-selected and likely had a strong interest in school food. Some regions and stakeholder groups were better represented than others and Wales was under-represented throughout the priority setting process. Other recruitment options could have been used, for example, quota sampling to ensure better representativeness, but these early stages were conducted in collaboration with other actors in the school food system, with the survey being conducted for a variety of purposes and the method of sharing was what was considered feasible at that time. Some workshop participants commented that parts of the workshop felt rushed, discussing the questions was repetitive and that the wording of the research questions was predetermined and could not be substantially amended. During the priority setting process, no differentiation was made between priorities for primary or secondary schools; some priorities may be more relevant to one setting than another and the detail of the priorities may differ. The broad perspective was taken pragmatically but may not then have fully captured the complexity of the system and the research needs, thus reducing their usefulness; but that broad perspective can also be considered a strength, with the detail being captured later as research questions are developed. Research priority setting will require periodic updating, and, at that point, consideration of the different school types and implications for priorities could be included.

## CONCLUSION

The priority setting exercise co-produced a set of the top 10 research priorities for school food-related research

encompassing subsidisation of school meals, implementation of policy and food standards, leadership, inequalities, social norms and healthy eating. These priorities, presented in the form of research questions, will help inform stakeholders, including research funders, to identify and conduct research and evaluate policies that have the greatest potential for change to the school food system.

**Acknowledgements** We would like to thank and acknowledge all participants who completed the surveys and took part in the online workshop as well as Toto Grunland who provided guidance on the JLA priority setting process and all other JLA facilitators who helped run the priority setting workshop. We would also like to thank School Food Matters, School Food Review, the GENIUS network, Myles Bremner, Abigail Page, Stephanie Slater, Angela Mullan, Jayne Jones, Danielle McCarthy and Judith Hanvey for their support with this research.

**Collaborators** GENIUS network: DS, SS, NOK, CCC, DO, MCM, JVW.

**Contributors** JVW and MCM conceived the idea for the study. JVW, with input from NOK, developed the first online survey and JVW and DS developed the second online survey. DS analysed the findings from the first survey, primary schools; DO confirmed that the identified questions or priorities were also relevant and comprehensive based on the responses related to secondary schools and JVW devised the research questions from that analysis. CCC conducted the scoping of the literature that also fed into the initial priority list. DS analysed the findings from the second survey. DS and JVW organised the online workshop, with input from NOK. JVW, MCM and DS decided on any final changes to the research questions after the workshop. DS wrote the first draft of the manuscript and JVW led the editing of the manuscript. All authors revised the manuscript critically for intellectual content and gave final approval of the submitted and revised manuscript. The corresponding author (JVW) is the guarantor of this study and attests that all listed authors meet authorship criteria and that no others meeting the criteria have been omitted.

**Funding** This work was supported by the UK Prevention Research Partnership (MR/S03756X/1), which is funded by the British Heart Foundation, Cancer Research UK, Chief Scientist Office of the Scottish Government Health and Social Care Directorates, Engineering and Physical Sciences Research Council, Economic and Social Research Council, Health and Social Care Research and Development Division (Welsh Government), Medical Research Council, National Institute for Health Research, Natural Environment Research Council, Public Health Agency (Northern Ireland), The Health Foundation and Wellcome.

**Competing interests** None declared.

**Patient and public involvement** Patients and/or the public were involved in the design, or conduct, or reporting, or dissemination plans of this research. Refer to the 'Methods' section for further details.

**Patient consent for publication** Not applicable.

**Ethics approval** This study was approved by Faculty of Medicine Health and Life Sciences Research Ethics Committee, Queen's University Belfast (reference MHLS 22_87). Participants gave informed consent to participate in the study before taking part.

**Provenance and peer review** Not commissioned; externally peer reviewed.

**Data availability statement** Data are available on reasonable request. We are happy to share the original survey on request.

**ORCID iDs**
Désirée Schliemann http://orcid.org/0000-0002-8746-3002
Dilara Olgacher http://orcid.org/0000-0002-0175-9348

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
