## [Reviewer comments · BMJ Open]

ARTICLE DETAILS

TITLE (PROVISIONAL)	Identifying the top 10 research priorities for the school food system in the United Kingdom: a priority setting exercise
AUTHORS	Schliemann, Desiree; Spence, Suzanne; O’Kane, Niamh; Chiang, Cheng; Olgacher, Dilara; McKinley, Michelle C.; Woodside, Jayne; Network, GENIUS

VERSION 1 – REVIEW

REVIEWER	Jones, Matthew University of the West of England Bristol, Health and Applied Social Sciences
REVIEW RETURNED	15-Nov-2023

GENERAL COMMENTS	Identifying the top 10 research priorities for the school food system in the United Kingdom: a priority setting exercise Title – This is clear and a fair description of the study Abstract – Clear overall. I would have preferred more attention to summarising the 10 research priorities. These are results that will interest audiences, possibly more so than the execution of the JLA process. Background – This is clear and succinct. Methods - Quite complex to explain, but clearly justified and transparent overall. Ethical approval and the PPI aspect was flagged in different parts of the paper. Eg P9 Lines 181-883 and P14 Line 303 and elsewhere. I appreciate that the effort was to be accountable about the ethical aspects at each stage, however the references were a bit repetitive and might have been condensed into one summary. Results. Overall, these are clearly set out. The table of priorities appear well framed and useful, which is a good indicator of the quality of the processes underpinning the outcomes of the study. Discussion. A good account overall of the novelty of the study in relation to other work. Some discussion of the limitations. Were other recruitment options assessed, for instance a quota sampling approach for Survey 1? I am not convinced it was a good decision to include both primary and secondary schools within the frame of the study. It added to the complexity to the task of obtaining representative voices. The pronounced differences in issues between these settings makes it
--

	much harder to arrive well framed research questions. Overall, the broad perspective appears to have muddied the picture and reduced the usefulness of the results. However, I accept that this is a research choice and some rationale is given. I was curious to see that food education did not clearly feature appear in the final priorities. Is there anything the authors can report about this? Did it feature, but fail to reach the final 18? Could it have been side lined given attention to food provisioning questions? Is it implicit within some priorities (#7, #9)? Overall, the absence is noteworthy given its clear position as one aspect of the school food system and prominence in current agendas (see Dimpleby H. National Food Strategy: Independent Review. 2021). Conclusion. This is clear and succinct. Presentation – Good overall. See typos: P6 Line 93 surplus apostrophe after Genius network P19 Table ‘Parent/carer’ – capital for Carer for consistency
--	---

REVIEWER	Vine, Michelle M. Brock University, Health Sciences
REVIEW RETURNED	16-Nov-2023

GENERAL COMMENTS	Excellent paper. I appreciated the opportunity to review it, and have included minor revisions for the authors.  1. This paper requires an editorial review for grammar and punctuation. 2. I would recommend adding in some next steps at the end of the paper. The results are clear and concise, and flow directly from the priority setting exercise. It would be nice to see how the priorities could be implemented through next steps in research and evaluation.
---

VERSION 1 – AUTHOR RESPONSE

Reviewer:1 Comments to the Author:

- Title – This is clear and a fair description of the study

- Abstract – Clear overall. I would have preferred more attention to summarising the 10 research priorities. These are results that will interest audiences, possibly more so than the execution of the JLA process.

Author response: The abstract has been modified as suggested, although it is challenging given overall word limits.

- Background – This is clear and succinct.

- Methods - Quite complex to explain, but clearly justified and transparent overall.

- Ethical approval and the PPI aspect was flagged in different parts of the paper. Eg P9 Lines 181-883 and P14 Line 303 and elsewhere. I appreciate that the effort was to be accountable about the ethical aspects at each stage, however the references were a bit repetitive and might have been condensed

into one summary.

Author response: We have kept the declarations separate but have merged the various sections where Ethics and PPI were mentioned and placed them at the end of the various stage descriptions as requested for clarity.

- Results. Overall, these are clearly set out. The table of priorities appear well framed and useful, which is a good indicator of the quality of the processes underpinning the outcomes of the study.

- Discussion. A good account overall of the novelty of the study in relation to other work.

- Some discussion of the limitations. Were other recruitment options assessed, for instance a quota sampling approach for Survey 1?

Author response: This limitation and an explanation of it has been added (see lines 304-308)

- I am not convinced it was a good decision to include both primary and secondary schools within the frame of the study. It added to the complexity to the task of obtaining representative voices. The pronounced differences in issues between these settings makes it much harder to arrive well framed research questions. Overall, the broad perspective appears to have muddied the picture and reduced the usefulness of the results. However, I accept that this is a research choice and some rationale is given.

Author response: This limitation and an explanation of it has been added (although we believe that the broad perspective, as well as being pragmatic, can also be seen as a strength); see lines 304-308

- I was curious to see that food education did not clearly feature appear in the final priorities. Is there anything the authors can report about this? Did it feature, but fail to reach the final 18? Could it have been side lined given attention to food provisioning questions? Is it implicit within some priorities (#7, #9)? Overall, the absence is noteworthy given its clear position as one aspect of the school food system and prominence in current agendas (see Dimpleby H. National Food Strategy: Independent Review. 2021).

Author response: Food education was considered within priority 7, and this is now described within the discussion – but it certainly did not feature at any stage as strongly as the food provision questions (line 276-279).

- Conclusion. This is clear and succinct.

- Presentation – Good overall. See typos:

P6 Line 93 surplus apostrophe after Genius network

P19 Table 'Parent/carer' – capital for Carer for consistency

Author response: These changes have been completed.

- Competing interests of Reviewer: None

Reviewer: 2 Comments to the Author:

- Excellent paper. I appreciated the opportunity to review it, and have included minor revisions for the authors.

- This paper requires an editorial review for grammar and punctuation.

Author response: This has been completed.

• I would recommend adding in some next steps at the end of the paper. The results are clear and concise, and flow directly from the priority setting exercise. It would be nice to see how the priorities could be implemented through next steps in research and evaluation.

Author response: We believe this exists to some extent in lines 286-296, but we have added some extra detail to this section; we have also added some further ideas around future steps in response to reviewer 1 e.g. lines 314-317.

• Competing interests of Reviewer: No competing interests.

Kind regards,
Prof Jayne V. Woodside

VERSION 2 – REVIEW

REVIEWER	Vine, Michelle M. Brock University, Health Sciences
REVIEW RETURNED	01-Jan-2024
GENERAL COMMENTS	Excellent job on the revisions of this paper. I would strongly suggest another read through for grammar and punctuation. I look forward to seeing this paper in print!